# Improved Electrophoretic Separation to Assist the Monitoring of Bcl-xL Post-Translational Modifications

**DOI:** 10.3390/ijms20225571

**Published:** 2019-11-08

**Authors:** Claude Bobo, Claire Céré, Mélody Dufossée, Alain Dautant, Violaine Moreau, Stéphen Manon, Florian Beaumatin, Muriel Priault

**Affiliations:** 1CNRS, Institut de Biochimie et de Génétique Cellulaires, UMR 5095, 1 Rue Camille Saint-Saëns, 33077 Bordeaux, France; bobo@ibgc.cnrs.fr (C.B.); claire.cere@crpp.cnrs.fr (C.C.); a.dautant@ibgc.cnrs.fr (A.D.); manon@ibgc.cnrs.fr (S.M.); beaumatin@ibgc.cnrs.fr (F.B.); 2Université de Bordeaux, Institut de Biochimie et de Génétique Cellulaires, UMR5095, 1 Rue Camille Saint-Saëns, 33077 Bordeaux, France; 3INSERM U1053, Bordeaux Research in Translational Oncology, 146, Rue Léo Saignat, 33076 Bordeaux, France; violaine.moreau@inserm.fr; 4Université de Bordeaux, 146, Rue Léo Saignat, 33076 Bordeaux, France

**Keywords:** Bcl-x_L_, post-translational modification, deamidation, phosphorylation, electrophoresis

## Abstract

Bcl-x_L_ is an oncogene of which the survival functions are finely tuned by post-translational modifications (PTM). Within the Bcl-2 family of proteins, Bcl-x_L_ shows unique eligibility to deamidation, a time-related spontaneous reaction. Deamidation is still a largely overlooked PTM due to a lack of easy techniques to monitor Asn→Asp/IsoAsp conversions or Glu→Gln conversions. Being able to detect PTMs is essential to achieve a comprehensive description of all the regulatory mechanisms and functions a protein can carry out. Here, we report a gel composition improving the electrophoretic separation of deamidated forms of Bcl-x_L_ generated either by mutagenesis or by alkaline treatment. Importantly, this new gel formulation proved efficient to provide the long-sought evidence that even doubly-deamidated Bcl-x_L_ remains eligible for regulation by phosphorylation.

## 1. Introduction

The oncogene Bcl-x_L_ is an anti-apoptotic protein of the Bcl-2 family that essentially functions to promote cell survival. Within the Bcl-2 family of proteins, Bcl-x_L_ achieves apoptosis inhibition by engaging Bcl-2 Homology motives (BH1, 2, and 3) in canonical interactions to sequester the BH3 motif of pro-apoptotic members. Outside of the Bcl-2 family of proteins, Bcl-x_L_ engages other binding partners, such as, for example, VDAC1 [1], p53 [2], or RAS [3], mostly through its BH4 domain but not only, and thus regulates other survival-related functions, such as calcium homeostasis or stemness. More recently, the unstructured loop that links BH4 to BH1 in Bcl-x_L_ was also shown to mediate the allosteric regulation [4] previously proposed to be triggered by p53 to facilitate Bcl-x_L_ interactions with BH3 only proteins [5]. Remarkably, all of these interactions are regulated by post-translational modifications.

Post-translational modifications (PTMs) allow cells to finely tune temporal expression and spatial distribution of proteins, and they are a means by which they shape their interactome, and thus modulate their functions in response to various cues. Being able to detect PTMs is essential to not only achieve a comprehensive description of all independent regulatory mechanisms and functions a protein can carry out but also to investigate possible cross-talks between different signaling pathways and establish a hierarchy in the PTMs a given protein can accumulate. Bcl-x_L_ is known to undergo different types of PTMs, most of them occurring in its unstructured loop: phosphorylation in response to microtubule inhibition, caspase cleavage upon an apoptosis trigger, ubiquitylation, and deamidation in response to DNA damage (for review see [6]). Among these, deamidation is probably the less studied due to a lack of simple techniques to accurately monitor this PTM without the need for specialized equipment. Indeed, deamidation converts asparagine residues into aspartate/isoaspartate and glutamine residues into glutamate. Each deamidation only confers a mass increase of 1 Da and an additional negative charge. Mass spectrometry is considered best suited to characterize these changes, with the caveat that sample preparation involves steps that can accelerate deamidation. Deamidation remains largely inconspicuous when proteins are separated by standard tris–glycine SDS-PAGE (commonly referred to as Laemmli gels). As a result, for more than ten years, Bcl-x_L_ was essentially described as a doubly deamidated protein on N52 and N66. Only recently did the use of 25 cm-long SDS-PAGE allow us to emphasize the existence of N52 monodeamidated Bcl-x_L_, thus pinpointing that N52 and N66 were not equally susceptible to deamidation and that a sequential mechanism was actually at play, always involving N52 first [7].

We set out to improve the separation of deamidated species and to achieve with polyacrylamide mini-gels the same resolution as that obtained with 25 cm-long tris–glycine PAGE while decreasing the time of electrophoresis. We report here that using taurine or asparagine as trailing ions proved efficient to increase the migration shifts caused by deamidation. Finally, in a previous study asking whether Bcl-x_L_ could accumulate both deamidation and phosphorylation, we had provided compelling evidence that monodeamidated Bcl-x_L_ could be phosphorylated in response to microtubule inhibition. However, we and others had failed to determine whether doubly-deamidated Bcl-x_L_ could be phosphorylated because of insufficient electrophoretic resolution provided by tris–glycine PAGE [7,8]. We show here that taurine–glycine gels proved efficient to overcome this limitation and found that doubly-deamidated Bcl-x_L_ is eligible for regulation by phosphorylation. 

## 2. Results

### 2.1. Separation of Bcl-x_L_ Deamido-Mimetic Mutants

Bcl-x_L_ undergoes deamidation on Asn52 and Asn66, and mutagenesis on these two positions can be used either to recapitulate deamidation by Asn→Asp conversion or to prevent it by Asn→Ala conversion. The migration distances of these mutants, listed in Figure 1, have previously been compared by tris–glycine PAGE to determine why Bcl-x_L_ migrates as a doublet in all the cell extracts we analyzed [7], and to identify the PTM responsible for the slower migration of a fraction of cellular Bcl-x_L_ (labeled * in WT lanes of gels in Figure 1) [7]. Long 25 cm tris–glycine SDS-PAGE was needed (Figure 1a) to obtain a resolution that 6 cm mini-gels did not allow (Figure 1b). Based on previous reports showing that taurine (i.e., 2-aminoethanesulfonate) can be used in SDS-PAGE as an alternative trailing ion to improve the resolution of small proteins [9], we tested three different discontinuous 6 cm-mini gel compositions to separate the deamido-mimetic mutants of Bcl-x_L_. Chloride was used as front ions, and trailing ions were either taurine–glycine (Figure 1c) or taurine–asparagine (Figure 1d) or asparagine–glycine (Figure 1e). In all three cases, the stacking effect on proteins allowed a better separation of the deamido-mimetic mutants of Bcl-x_L_ than mini tris–glycine SDS-PAGE (Figure 1b). Importantly, while 30 h of migration was needed with the 25 cm-long gels, these new gel compositions thus allowed to cut it down to 1 h 30. Taurine–glycine gels accentuate the serendipitous “migration artifacts” allowing the detection of the additional 1 q- and increase of 1 Da per deamidation reaction, which otherwise remain hardly detectable.

Of note, modifying the trailing ions also allowed the discrimination of mutants based on their global hydrophobicity. Indeed, the double substitution of N52 and N66 by alanine residues allowed the Bcl-x_L_ N52AN66A mutant to migrate further than the unmodified protein in the gel (compare lane “AA” to lane “WT” on gels c, d, and e in Figure 1). In addition, one single Asn→Ala substitution generated a shift in the migration distance (compare lanes “AD” and “N66D”, and compare lanes “N52D” and “DA” on gels c, d, and e in Figure 1). The fine separation of proteins granted by these new gel compositions can indeed be used to accurately pinpoint that the modified form (labeled * in WT lanes of Figure 1) above unmodified Bcl-x_L_ co-migrates exactly with the single mutant Bcl-x_L_ N52D, thus confirming what we previously reported with long tris–glycine PAGE [7]. However, these gels still show a difference in the migration distances of Bcl-x_L_ AD and Bcl-x_L_ DA (as 25 cm-long tris–glycine SDS-PAGE already did) even though both proteins have the exact same molecular weight. Electrophoresis is known to separate proteins according to their apparent MW, rather than their absolute MW. We can only surmise that different structural constraint(s) that is (are) not alleviated by denaturing conditions contribute different migration properties that account for differences in the apparent molecular weight of Bcl-x_L_ AD and Bcl-x_L_ DA.

### 2.2. Separation of Chemically-Induced Deamidated Bcl-x_L_

Protein folding is an integral component of migration distances in electrophoresis, even under denaturing conditions. IsoAsp formation is known to have profound structural aftermaths on deamidated species because the protein backbone is rerouted to incorporate a supplementary carbon atom [6]. In Bcl-x_L_, even though N52 and N66 are located in an unstructured loop, any structural reorganization might account for the gel retardation of deamidated species in taurine–glycine gels, so we asked whether migration distances in Figure 1d could be a telltale of IsoAsp formation. Extreme pHs, both alkaline and acidic, are known to increase the probability of deamidation of eligible Asn residues; the reaction mechanisms and rates are different and respectively lead to Asp/IsoAsp mixture or only Asp [10,11]. Therefore, recombinant human Bcl-x_L_ was exposed to alkaline pH in conditions that were previously shown to produce extensive double deamidation of Bcl-x_L_ (see Figure 1 in [7]). Before any electrophoretic separation, we ensured that the protracted alkaline treatment had not caused any dramatic loss in Bcl-x_L_ structure, which would invalidate our demonstration. Circular dichroism spectra before and after alkaline treatment showed that the protein retained the same fold, with maxima and minima typical of a protein composed essentially of alpha-helices (Figure 2a). Calculated secondary structures indicated a slight decrease in helix content after chemical deamidation (33.8% vs. 26.2%) (Appendix A). Studying the folding of Bcl-x_L_ as a function of temperature actually revealed Bcl-x_L_ as a strikingly thermostable protein (Figure 2b). We then proceeded with separation on taurine–glycine gels (Figure 2c): Untreated Bcl-x_L_ appeared as a doublet consisting of unmodified and N52 monodeamidated species. The product(s) generated after alkaline treatment showed as a single band with a migration distance corresponding to that observed in Figure 1 when Bcl-x_L_ double deamidation resulted from Asp substitution of Asn52 and Asn66. We concluded that alkaline treatment of the recombinant protein proved efficient to accelerate Bcl-x_L_ deamidation, and did not reveal any other deamidated site in the protein. Moreover, separation on taurine–glycine gels did not allow the discrimination between Asp and IsoAsp deamidated forms.

### 2.3. Detection of Accumulated PTMs.

In a previous study, we asked whether deamidated Bcl-x_L_ retained the ability to be modified by phosphorylation in response to microtubule inhibition. We could only partly answer the question: The mutant recapitulating monodeamidation Bcl-x_L_ N52DN66A was indeed phosphorylated in vinblastine treated cells, but the mutant recapitulating double-deamidation Bcl-x_L_N52DN66D did not show any additional migration shift on tris–glycine gels in response to vinblastine [7]. We repeated these experiments with taurine–glycine gels this time, and the resolution granted allowed us to detect an additional migration shift in vinblastine-treated cells expressing Bcl-x_L_N52DN66D. This slower migrating band disappeared in λ-phosphatase-treated samples (Figure 3). We thus showed for the first time that the deamido-mimetic mutant Bcl-x_L_N52DN66D could accumulate an additional PTM: namely phosphorylation.

## 3. Discussion

### 3.1. Assisting the Detection of Deamidation

We reported here how replacing or mixing low molecular weight trailing ions, such as glycine, with larger ones, such as taurine or asparagine, in denaturing polyacrylamide gels composition greatly assists the electrophoretic separation of the deamidated and phosphorylated forms of Bcl-x_L_. Compared to tris–glycine SDS-PAGE, one major drawback is the short shelf-life of the stock solutions, which need to be renewed monthly to grant proper results. But this drawback is largely overcome by the improvement achieved with bands separation. The resolution was such that (1) slight modifications in the hydrophobicity of the protein triggered gel retardation when Asn residues were substituted by Ala, and (2) even a single deamidation reaction could be monitored. Given that the additional negative charge and increase by 1 Da imparted by each deamidation reaction should, in theory, remain hardly detectable by electrophoretic separation, it is justified to say that taurine–glycine gels accentuate the “migration artifacts” at play with Bcl-x_L_ deamidated forms. The potent structural alteration conveyed by IsoAsp residues could have been invoked in the process, but we ruled out this hypothesis by showing that taurine–glycine gels did not allow discrimination between Asp and IsoAsp deamidated forms. Whether this easy and costless technique can be transposed to other deamidated proteins, and whether Gln deamidation can be detected as well will require a case by case analysis. Ditto for the separation of other PTMs. 

### 3.2. Evidence for Accumulation of PTMs by Bcl-x_L_

Depending on external cues, cells trigger different and independent pathways to convey a signal and implement the appropriate response. In this regard, Bcl-x_L_ anti-apoptotic functions can not only be regulated by different PTMs depending on the stimulus applied but also depending on the cell type. Many cell types trigger Bcl-x_L_ phosphorylation on Ser62 in response to microtubule inhibition [8,12], while Bcl-x_L_ double deamidation is observed in response to DNA-damage (for review [6]). Whether these independent signaling pathways can act in concert to produce a protein accumulating both PTMs was previously addressed but only partly answered: Tris–glycine gels only allowed us to demonstrate that mono-deamidated Bcl-x_L_ could still be a substrate for phosphorylation after vinblastine treatment, but no conclusion could be drawn for doubly-deamidated Bcl-x_L_ [7]. In keeping, when Upreti et al. mapped the site of phosphorylation to Ser-62, their mass spectrometry analysis revealed complete deamidation of Asn52 and 66, but the authors could not rule out that deamidation had occurred during sample preparation (alkaline elution of immunoprecipitated Bcl-x_L_) [8]. The new gel composition reported in this study finally allowed us to show that doubly-deamidated Bcl-x_L_ remains a substrate for phosphorylation upon microtubule inhibition. According to the literature, Ser62 is the phosphorylated residue. We can thus infer that the presence of two additional negative charges in close vicinity (positions 52 and 66) does not alter recognition by the kinase for subsequent phosphorylation. This is an interesting demonstration that Bcl-x_L_ could accumulate PTMs if the respective pathways were to intersect.

### 3.3. Deamidation of Bcl-x_L_: Open Questions and Applications

Within the Bcl-2 family of proteins, Bcl-x_L_ stands out by virtue of several intrinsic (rather than functional) traits: Its unique eligibility to deamidation is one of them, and its remarkable stability is yet another. We showed here that purified Bcl-x_L_ remains folded even after protracted incubation in extreme conditions (pH = 10 for 24 h at 37 °C) that lead to extensive double deamidation. By comparison, the related pro-apoptotic protein Bax is unfolded and degraded within the first hours of the same treatment (our unpublished data). The stability of purified Bcl-x_L_
*in vitro* echoes with the cellular studies we performed, and which revealed that Bcl-x_L_ (deamidated or not) remains stable for over a day in cells [7]. Such a trait makes Bcl-x_L_ a most suitable model to study a time-related slow process such as deamidation both *in vitro* and in a cellular context.

Working on the electrophoretic separation of Bcl-x_L_, we discovered its obligatory sequential deamidation of N52 first, and then of N66 [7], thus providing experimental demonstration of the accuracy of the NGOME algorithm that computes sequence-derived secondary structure and intrinsic disorder parameter, and calculates a shorter half-life for N52 than for N66 [13]. Recent structural data further underpin the sequential mechanism with NMR structures showing that N52 is more frequently exposed to solvent, while N66 is kept close to the folded core of the protein (2ME9 PDB file) [4]. Helped by the new gel composition described here, the exciting avenue is now opened to achieve a full functional characterization of N52 deamidated Bcl-x_L_. We previously discovered that monodeamidation is not merely a bystander modification, but that it alters Bcl-x_L_ oncogenic and tumorogenic properties [7]. Therefore, one last expected outcome of the present work is that monitoring Bcl-x_L_ deamidation profile in patients might also assist diagnose pathologies related to Bcl-x_L_ dysfunctions.

## 4. Materials and Methods 

### 4.1. Recombinant Proteins

Recombinant Bcl-x_L_ was expressed in BL21 *Escherichia coli* strain as an N-terminal fusion protein with a 10His tag, and a linker containing a Factor Xa cleavage site. After affinity-purification, as described in [14], the protein was stored in PBS.

### 4.2. Electrophoresis

Taurine–glycine resolving gel composition: 75 mM tris (pH = 10) (Euromedex, Strasbourg, France), 200 mM taurine, 125 mM glycine (Euromedex, Strasbourg, France), 23 mM HCl, 12% acryl/bisacryl 37.5:1 (Biosolve, Dieuze, France). Taurine–asparagine resolving gel composition: 75 mM tris–HCl (pH = 8.8, ), 100 mM taurine, 100 mM asparagine (Sigma-Aldrich, St. Louis, MO, USA), 12% acryl/bisacryl 37.5:1. Glycine–asparagine resolving gel composition: 75 mM tris–HCl (pH = 8.8), 125 mM glycine, 150 mM asparagine, 23 mM HCl, 12% acryl/bisacryl 37.5:1. SDS-PAGE (tris–glycine) resolving gels composition: 380 mM tris–HCl pH = 8.8, 0.1% SDS (Sigma-Aldrich), 12% acryl/bisacryl 37.5:1. Gel polymerization was induced by the addition of APS (0.6 mg/mL, Sigma-Aldrich) and Temed (1 µL/mL, Sigma-Aldrich). Stacking mini-gels formulation: 125 mM tris–HCl pH = 6.7 and 5% acryl/bisacryl 37.5:1. In the case of SDS-PAGE, 0.1% SDS was added. Migration buffer for all the electrophoretic separations was glycine 14.4 g/L, tris 3 g/L, SDS 1 g/L, pH = 8.8. Electrophoretic conditions are set at 25 mA per mini-gel.

### 4.3. Circular Dichroism

CD spectra were recorded in the far-UV region with a JASCO J810 spectropolarimeter (JASCO Corporation, Tokyo, Japan). For each experiment, 0.15 mg/mL proteins in 10 mM phosphate buffer (pH = 7.4) were placed in a 0.1 cm quartz cuvette and monitored in continuous scan mode. Data were averaged form 6 scans.

Folding of Bcl-x_L_ was also followed as a function of temperature, with increases of 1 °C/min.

Analysis for protein CD spectra was performed with DichroWeb [15,16] to compare untreated and deamidated Bcl-x_L_: The values were analyzed with the SELCON3 program (the self-consistent method) using the SP175 reference dataset [17].

### 4.4. Alkaline Treatment

Protein samples were incubated with 25 mM glycine-NaOH, pH = 10 for 24 h at 37 °C. When needed, samples were neutralized with HCl.

### 4.5. Cell Lines and Culture

HCT116 cells expressing Bcl-x_L_ deamidation mutants were described in [7]. When needed, endogenous Bcl-x_L_ was silenced by lentiviral infection, as described in [18]. Vinblastine (100 nM, MP Biomedical, Illkirch, France) was applied for 32 h.

### 4.6. Total Proteins Extraction

Cells were harvested and the pellets were resuspended in a RIPA buffer (100 mM tris–HCl (pH = 7.4), 0.5% NP-40 (Fisher scientific, Illkirch, France), 0.5% sodium-deoxycholate (Sigma-Aldrich), 0.1% SDS supplemented with proteases inhibitor Mini^®^ (Roche Diagnostics, Basel, Switzerland)) for 20 min. The solubilized proteins were then recovered in the supernatant after a 20 min-centrifugation at 12,000× *g*.

### 4.7. Lambda Phosphatase Treatment

When cells treated with vinblastine were harvested, the pellets were separated in two: one where total proteins were extracted in the presence of phosphatase inhibitors (10 mM NaF, 1 mM Na_3_VO_4_, 1 mM phenylmethanesulfonylfluoride, Sigma-Aldrich), and one where phosphatase inhibitors were omitted. The latter extracts were submitted to dephosphorylation with 200 units of λ-phosphatase (#P0753S, Biolabs, Ipswich, MA, USA) for 2 h at 37 °C. The reaction was stopped by 4% SDS, 125 mM tris pH = 6.8, 20% glycerol (Sigma-Aldrich), 0.002% (*w*/*v*) bromophenol blue (Sigma-Aldrich).

### 4.8. Western Blots

Proteins separated on polyacrylamide gels were transferred onto nitrocellulose blotting membranes (Amersham Protran™ 0,2 μm NC, Chicago, IL, USA)), and Western blots revealed with Clarity Western ECL (Bio-Rad laboratories, Marnes-la-Coquette, France).

Blocking solution was 3% nonfat milk.

Antibodies used were rabbit anti-Bcl-x (ab32370, Abcam, Cambridge, UK)). Horseradish peroxidase-conjugated secondary antibodies were from Jackson Immunoresearch (Cambridgeshire, UK).

Chemiluminescence was detected with a G:Box imaging system (Syngene, Cambridge, UK)).

## Figures and Tables

**Figure 1 ijms-20-05571-f001:**
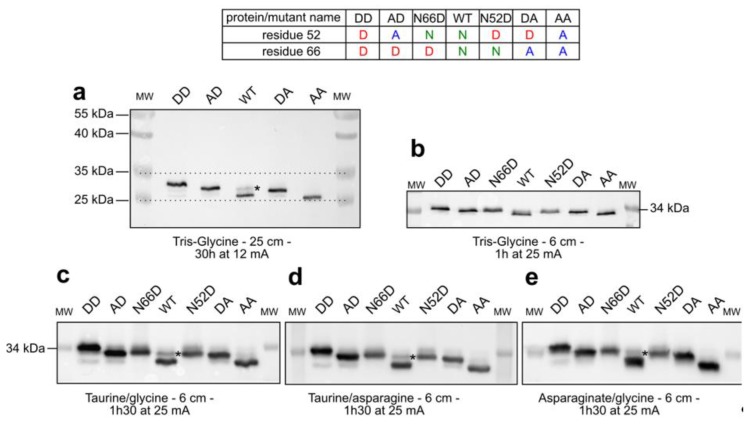
Separation of deamidated mutants of Bcl-x_L_ on different polyacrylamide gels: Total proteins were extracted from HCT116 expressing the indicated forms of Bcl-x_L_, and 25 µg were separated by standard tris–glycine PAGE on a 25 cm-long gel for 30 h (**a**) or 6 cm-mini gels of the indicated compositions for 1 h 30 min (**b**–**e**). * indicates the modified form of Bcl-x_L_ identified as N52-monodeamidated Bcl-x_L_ according to migration distances.

**Figure 2 ijms-20-05571-f002:**
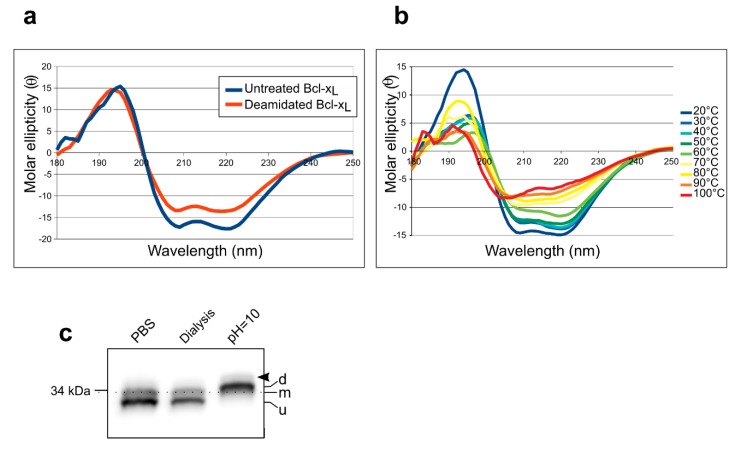
Folding and electrophoretic separation of chemically-induced deamidated Bcl-x_L_: (**a**) Storage buffer for purified His10-Bcl-x_L_ (PBS) was replaced by dialysis with 10 mM phosphate (pH = 7.4) to accommodate circular dichroism requirements, and spectra were recorded between 180 and 250 nm on the untreated protein, or after alkaline treatment to accelerate deamidation. (**b**) Thermic denaturation of His10-Bcl-x_L_ in 10 mM phosphate (pH = 7.4). (**c**) Immunodetection of Bcl-x_L_ after separation on taurine–glycine gels of 25 µg of purified His10-Bcl-x_L_ stored in PBS, or after dialysis against 10 mM phosphate, and after alkaline treatment. U: unmodified, m: monodeamidated, d: doubly-deamidated. Note that the linker between His10 and Bcl-x_L_ contains an asparagine residue that introduces an additional deamidation site outside of Bcl-x_L_ (arrow head).

**Figure 3 ijms-20-05571-f003:**
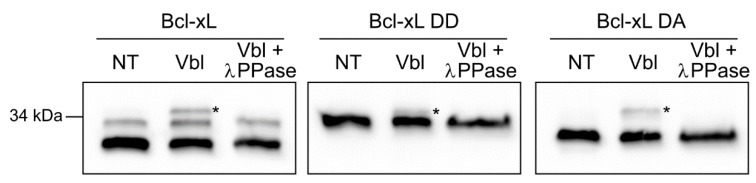
Electrophoretic separation of deamidated and phosphorylated proteins: HCT116 cells expressing the indicated forms of Bcl-x_L_ were infected with lentiviruses to silence the expression of endogenous Bcl-x_L_ (the genes encoding the proteins re-expressed are modified so that their mRNA is not targeted by the ShRNA used to silence endogenous Bcl-x_L_). Cells were either left untreated or exposed to Vinblastine (100 nM) for 32 h to induce Bcl-x_L_ phosphorylation. Total proteins were extracted and, where indicated, were treated with λ-phosphatase. Twenty micrograms of proteins were separated on taurine–glycine mini-gels, and immunodetection of Bcl-x_L_ was performed.

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
