# Peer review of "Improved Electrophoretic Separation to Assist the Monitoring of Bcl-xL Post-Translational Modifications"

_ijms, 2019, doi:10.3390/ijms20225571_

Round 1

Reviewer 1 Report

The ms by Bobo and colleagues is a short communication reporting a new method regarding the electrophoretic separation of deaminated forms of the apoptosis inhibitor Bcl-xL. This work is the continuation of their previous work published in Oncotarget about the oncogenic properties of deaminated Bcl-xL.

Here the author have set up a new SDS PAGE formulation that improves the separation of deaminated Bcl-xL forms and significantly shortens electrophoresis duration (from 30h down to 1h30). In addition, the authors show a series of data showing that this new protocol may apply to other post-translation modifications beyond Bcl-xL.

Overall this work is potentially of interest for a wide range of readers.  

Some issues to be adressed by the authors are listed below.

N66D and N52D mutants are lacking in Fig 1a. Apparently N66D and N52D have distinct apparent MW, this should be discussed. According to Fig 1a, the apparent MW of DD mutant seems to be around 33-32kDa, whereas in Fig 1b it is higher than 34kDa. This should be clarified. In this regard, the author should show entire minigels, not only a fraction. Figs 1c-e. Overall the bands appear to be poorly focalised compared to Figs 1 a & b. This issue should be discussed. Fig 2a. Deamination seems to induce denaturation to some extent. Percentage of alpha helix should be indicated. The authors may whish to play with DMF concentration regarding this issue. Fig 2b. Same remark as above Figs 3b-d. The authors should show the entire minigels, including MW ladders. Fig 3d. Right panel, Ctrl pattern is reminiscent of protein degradation. This issue should be secured.

Author Response

Reviewer 1:

The ms by Bobo and colleagues is a short communication reporting a new method regarding the electrophoretic separation of deaminated forms of the apoptosis inhibitor Bcl-xL. This work is the continuation of their previous work published in Oncotarget about the oncogenic properties of deaminated Bcl-xL.

Here the author have set up a new SDS PAGE formulation that improves the separation of deaminated Bcl-xL forms and significantly shortens electrophoresis duration (from 30h down to 1h30). In addition, the authors show a series of data showing that this new protocol may apply to other post-translation modifications beyond Bcl-xL.

Overall this work is potentially of interest for a wide range of readers.  

Some issues to be adressed by the authors are listed below.

N66D and N52D mutants are lacking in Fig 1a.

Figure 1a shows the state of the art as it was after our publication in Oncotarget in 2016. At that time, we focused on the discovery of the monodeamidated form of Bcl-xL, and we concluded from the separation obtained with 25 cm-long Tris-glycine gels that monodeamidation occurred on position N52. It is only when we implemented Taurine/glycine gels that we realized that using Alanine in position 52 or 66 as a non-deamidable residue also changed the distance of migration, by virtue of it hydrophobic property. Combining the N52D and N66A mutations thus generated a mutant with a migration distance that did not exactly co-migrate with the modified band of Bcl-xL we were characterizing. Therefore, we needed to analyze the single mutants N66D and N52D on our new gels. These single mutants provided the disambiguation needed to ascertain our demonstration that the modification was the monodeamidation of N52. This is what we tried to stress lines 106-116.

Apparently N66D and N52D have distinct apparent MW, this should be discussed.

The formulation of this reviewer is absolutely correct: electrophoresis is not suited to determine “absolute MW”, but only “apparent MW”. We discussed this point as requested lines 116-123.

According to Fig 1a, the apparent MW of DD mutant seems to be around 33-32kDa, whereas in Fig 1b it is higher than 34kDa. This should be clarified.

The pre-stained MW standards we used were the same brand in all the experiments. It is very likely that the apparent molecular weights are a function of the migration distances, which is different in Figure 1a (25 cm) and Figures 1c-e (6 cm).

In this regard, the author should show entire minigels, not only a fraction.

We provided the full size images of each gel of the manuscript, as requested by the editors. We assume they will be available for anyone to see on the website or as supplementary data?

Figs 1c-e. Overall the bands appear to be poorly focalised compared to Figs 1 a & b. This issue should be discussed.

We agree with the reviewer’s comment. Figs 1a and b are standard Tris-Glycine gels, and they seem to show better focalised bands. The other gel compositions in Figs c-e provide better resolution, but likely at the expense of focalisation. This might be overcome by loading less proteins. Because we have no explanation or definite proof, we would rather not open a discussion that we cannot sustain.

Fig 2a. Deamination seems to induce denaturation to some extent. Percentage of alpha helix should be indicated. The authors may whish to play with DMF concentration regarding this issue.

The manuscript was modified accordingly (lines 141-143). We thank the reviewer for the advice regarding DMF for future work.

Fig 2b. Same remark as above Figs 3b-d. The authors should show the entire minigels, including MW ladders.

As previously said, we provided the full size images of each gel of the manuscript.

Fig 3d. Right panel, Ctrl pattern is reminiscent of protein degradation. This issue should be secured.

The exact same samples were used for analysis at the same time on the two types of gels in parallel, which is why we do not think that one panel in Figure 3d reflects protein degradation while the other does not. 4EBP1 is known to be poly-phosphorylated under control conditions (T37, T46, S65 and T70 are phosphorylated when mTOR is active), and what we claim is that Taurine/Glycine gels simply provide a different separation of the polyphorphorylated forms of 4EBP1.

In addition, the fact that the most intense signal is observed as the furthest migrating species seems to refute protein degradation, which would result in blurry lower migration bands.

Reviewer 2 Report

The authors have improved the separation of deamidated protein in polyacrylamide mini-gels. The authors have reported that taurine or asparagine are trailing ions to efficiently increase the protein migration shifts caused by deamidations, and exemplified by Bcl-XL, RhoA, and calmodulin. These results are also very interesting for characterization of the other PTMs. However, the manuscript suffers from several weaknesses:

Is it different between N52-monodeamidated Bcl-XL and N66-monodeamidated Bcl-XL? The authors are encouraged to show the quantification of Bcl-XL deamidation. What’s the effect of acid pH on protein deamidation? There are different ph conditions: pH 10 and PBS in Fig 2, pH 6.7 and 11 in Fig 3. The authors should explain this. The PTMs of the samples are encouraged to be confirmed by LC-MS before the gel analysis. Some gel bands are unclear.

Author Response

Reviewer 2:

The authors have improved the separation of deamidated protein in polyacrylamide mini-gels. The authors have reported that taurine or asparagine are trailing ions to efficiently increase the protein migration shifts caused by deamidations, and exemplified by Bcl-XL, RhoA, and calmodulin. These results are also very interesting for characterization of the other PTMs. However, the manuscript suffers from several weaknesses:

Is it different between N52-monodeamidated Bcl-XL and N66-monodeamidated Bcl-XL?

We discussed this point lines 116-123.

The authors are encouraged to show the quantification of Bcl-XL deamidation.

Our previous paper published in Oncotarget in 2016 already covered this topic, and we have no new data regarding this point. We could provide a quantification of deamidation when it is chemically induced (Figure 3) if the conversion was not total, which is not the case, so we do not feel this would add to the demonstration.

What’s the effect of acid pH on protein deamidation?

As mentioned in the initial version of the paper (lines 133-136 of the new manuscript) acid pH is known to induce protein deamidation through a different mechanism than alkaline pH, and at a different rate. “Extreme pHs, both alkaline and acidic, are known to increase the probability of deamidation of eligible Asn residues; the reaction mechanisms and rates are different and respectively lead to Asp/IsoAsp mixture or only Asp”. This is documented in refs [9,10].

There are different ph conditions: pH 10 and PBS in Fig 2, pH 6.7 and 11 in Fig 3. The authors should explain this.

We thank the referee for pointing out this mistake. We changed pH=11 to pH=10 (line 224 of the new manuscript).

The PTMs of the samples are encouraged to be confirmed by LC-MS before the gel analysis. Some gel bands are unclear.

The PTMs studied in Figure 3 have extensively been characterized by many different groups and reported in numerous articles in the literature; the migrations we monitor on classical Tris-Glycine gels (Laemmli gels) are in perfect agreement with these previously published results. This is why we felt confident to rely on the literature and did not feel the need to confirme the PTMs by mass spec analysis. We stress that the exact same samples were analyzed at the same time in parallel on Tris-Glycine and Taurine-Glycine gels. Therefore we simply highlight that Taurine/Glycine gels provide a different separation than Laemmli gels, and that this could potentially open new avenues for future studies.

Reviewer 3 Report

The manuscript by Bobo et al. describes an improved electrophoretic separation method for investigating deamidation status of proteins of interest, such as Bcl-xl. Based on its sufficiency, the method introduced here might be also interesting to the researchers who are focusing on the PTMs including lipidation and phosphorylation. However, this reviewer feels that the manuscript should have been carefully checked for its format and style before the submission. Some standards should be strictly followed in the whole manuscript. For example, leave a space between the numerical value and unit symbol.

Major points

1) The title should be revised to fit the manuscript since the method could be also be applied to other targets from other PTMs. 

2) The authors should also introduce other existing methods, such as "Phos-tag gel" for detecting phosphorylated POIs. This could not be avoided in discussion section.

3) More descriptions are required for other PTMs. It is insufficient for readers to follow the logic.

4) As for Fig. 3C and 3D, are there some explanations for different results between Tris-glycine and Taurine/glycine? More bands are noticed.

5) Some candidates for large POIs? The capacity of this methods should be explored and discussed.

Author Response

Reviewer 3

The manuscript by Bobo et al. describes an improved electrophoretic separation method for investigating deamidation status of proteins of interest, such as Bcl-xl. Based on its sufficiency, the method introduced here might be also interesting to the researchers who are focusing on the PTMs including lipidation and phosphorylation. However, this reviewer feels that the manuscript should have been carefully checked for its format and style before the submission. Some standards should be strictly followed in the whole manuscript. For example, leave a space between the numerical value and unit symbol.

We apologize for the inconvenience, and did our best to amend the new version according to the reviewer’s comment.

Major points

1) The title should be revised to fit the manuscript since the method could be also be applied to other targets from other PTMs.

We could propose “Improved electrophoretic separation to assist monitoring of post-translational modifications – focus on Bcl-xL deamidated forms”.

2) The authors should also introduce other existing methods, such as "Phos-tag gel" for detecting phosphorylated POIs. This could not be avoided in discussion section.

We modified the manuscript to accommodate the reviewer’s request (lines 267-276 of the new MS).

3) More descriptions are required for other PTMs. It is insufficient for readers to follow the logic.

We modified the manuscript to accommodate the reviewer’s request (lines 277-289 of the new MS).

4) As for Fig. 3C and 3D, are there some explanations for different results between Tris-glycine and Taurine/glycine? More bands are noticed.

A comment was added lines 256-260 of the new manuscript.

5) Some candidates for large POIs? The capacity of this methods should be explored and discussed.

We modified the discussion of the manuscript to accommodate the reviewer’s request (lines 277-291 of the new MS).

Round 2

Reviewer 2 Report

The revised manuscript has been improved from its original version, and the authors have addressed most of my concerns. There are some issues remain issues listed below:

The title is vague and non-specific. From the previous published paper in Oncotarget in 2016, the molecular weight of deamidation Bcl-xL is below 34 kDa (as shown as 1B). However, the current manuscript shows that it is 34 kDa (as shown as 1c-e). The authors should explain this. There are multi-PTMs in Bcl-xL, such serine/threonine phosphorylation and deamidation etc. Do other PTMs affect the electrophoretic separation of deamidated Bcl-xL? In Fig.2, was “deamidated Bcl-xL” monodeamidated or doublydeamidated? Do the other deamidated mutants of Bcl-xL still keep alpha-helices?

Author Response

The revised manuscript has been improved from its original version, and the authors have addressed most of my concerns. There are some issues remain issues listed below:

The title is vague and non-specific.

The title has been changed.

From the previous published paper in Oncotarget in 2016, the molecular weight of deamidation Bcl-xL is below 34 kDa (as shown as 1B). However, the current manuscript shows that it is 34 kDa (as shown as 1c-e). The authors should explain this.

This reviewer is correct. As a proof that our results are consistent, please note that Figure 1a of the current manuscript also shows that Bcl-xL migrates between the 25 kDa and 35 kDa markers, as in our Oncotarget paper. It thus appears that the length of the gels (25 cm in Figure 1a and in the Oncotarget paper vs 6 cm in Figures 1b-e) is responsible for a modification in the relative migrations of the MW standards and Bcl-xL species. As we stressed in our previous response to reviewer 1 “electrophoresis is not suited to determine “absolute MW”, but only “apparent MW”.

There are multi-PTMs in Bcl-xL, such serine/threonine phosphorylation and deamidation etc. Do other PTMs affect the electrophoretic separation of deamidated Bcl-xL?

This insightful comment allowed us to realized that we had already addressed this question in previous work, but could only answer partly. In our Oncotarget paper (Beaumatin et al 2016) we showed that monodeamidated Bcl-xL could still be phosphorylated in response to microtubule destabilizing agents (Fig 8), which triggered an additional mobility shift. But we could not conclude on the possibility for doubly deamidated Bcl-xL to be phosphorylated because we did not detect any mobility shift. Thanks to the reviewer’s question, we repeated the experiment with the new gel composition and could now show that even doubly deamidated Bcl-xL is eligible for phosphorylation. The manuscript was largely modified to accommodate these new data.

In Fig.2, was “deamidated Bcl-xL” monodeamidated or doublydeamidated?

As labelled on the figure, alkaline treatment of Bcl-xL results in complete double-deamidation. This was already shown in our Oncotarget paper (Figure 1). For more clarification, we amended the text of the current manuscript.

Do the other deamidated mutants of Bcl-xL still keep alpha-helices?

We did not analyze other deamidated mutants of Bcl-xL by circular dichroism or any other in vitro technique. However we did characterize deamidation mutants expressed in cells and found that they retain the same anti-apoptotic functions as unmodified Bcl-xL in terms of binding to pro-apoptotic partners and cell survival upon pro-apoptotic challenge (Beaumatin et al., Oncotarget 2016). This leads us to conclude that proteins maintain the alpha-helix fold regardless of the deamidation status.

Reviewer 3 Report

This revised manuscript resolved most of the comments by this reviewer. However, the reviewer feels that the authors should determine a focus of their study based on the results. A mass spec seems very essential to support the main results because multiple bands were observed. 

As for the title, is It really needed to include the subtitle?

Author Response

This revised manuscript resolved most of the comments by this reviewer. However, the reviewer feels that the authors should determine a focus of their study based on the results. A mass spec seems very essential to support the main results because multiple bands were observed.

This reviewer’s comment helped us realize that the manuscript as we wrote it was drifting away from the intended scope of the Special Issue. The scope of the IJMS Special Issue is on “recent advances on Bcl-xL”. We agree that mixing data on Bcl-xL and other proteins blurs the take-home message. Thanks to the comments raised by another reviewer (Reviewer 2), we could obtain more compelling data on Bcl-xL. We now propose a version of the article that is quite different from the previous one, but certainly fits better the scope of the Special Issue with a focus on Bcl-xL only. We hope Reviewer 3 will agree with the modifications brought.

As for the title, is It really needed to include the subtitle?

A new title was proposed.

Round 3

Reviewer 2 Report

The revised manuscript has been improved, and the authors have addressed all my concerns.